# Epigenetic Regulation of Estrogen Receptor Genes’ Expressions in Adipose Tissue in the Course of Obesity

**DOI:** 10.3390/ijms23115989

**Published:** 2022-05-26

**Authors:** Krzysztof Koźniewski, Michał Wąsowski, Marta Izabela Jonas, Wojciech Lisik, Maurycy Jonas, Artur Binda, Paweł Jaworski, Wiesław Tarnowski, Bartłomiej Noszczyk, Monika Puzianowska-Kuźnicka, Alina Kuryłowicz

**Affiliations:** 1Mossakowski Medical Research Center, Department of Human Epigenetics, Polish Academy of Sciences, 5 Pawinskiego St., 02-106 Warsaw, Poland; kkozniewski@imdik.pan.pl (K.K.); martajonas@imdik.pan.pl (M.I.J.); mpuzianowska@imdik.pan.pl (M.P.-K.); 2Medical Centre of Postgraduate Education, Department of General Medicine and Geriatric Cardiology, 231 Czerniakowska St., 00-401 Warsaw, Poland; mwasowski@cmkp.edu.pl; 3Department of General and Transplantation Surgery, The Medical University of Warsaw, 59 Nowogrodzka St., 02-014 Warsaw, Poland; wojciech.lisik@wum.edu.pl (W.L.); morjon@poczta.onet.pl (M.J.); 4Medical Center of Postgraduate Education, Department of General, Oncological and Bariatric Surgery, 231 Czerniakowska St., 00-401 Warsaw, Poland; artur.binda@interia.pl (A.B.); pjaworski@cmkp.edu.pl (P.J.); wtarnowski@cmkp.edu.pl (W.T.); 5Medical Center of Postgraduate Education, Department of Plastic Surgery, 231 Czerniakowska St., 00-401 Warsaw, Poland; noszczyk@melilot.pl; 6Medical Centre of Postgraduate Education, Department of Geriatrics and Gerontology, 61/63 Kleczewska St., 01-826 Warsaw, Poland

**Keywords:** obesity, adipose tissue, estrogen receptor, DNA methylation, microRNA interference

## Abstract

Estrogen affects adipose tissue function. Therefore, this study aimed at assessing changes in the transcriptional activity of estrogen receptor (ER) α and β genes (*ESR1* and *ESR2*, respectively) in the adipose tissues of obese individuals before and after weight loss and verifying whether epigenetic mechanisms were involved in this phenomenon. *ESR1* and *ESR2* mRNA and miRNA levels were evaluated using real-time PCR in visceral (VAT) and subcutaneous adipose tissue (SAT) of 78 obese (BMI > 40 kg/m^2^) and 31 normal-weight (BMI = 20–24.9 kg/m^2^) individuals and in 19 SAT samples from post-bariatric patients. *ESR1* and *ESR2* methylation status was studied using the methylation-sensitive digestion/real-time PCR method. Obesity was associated with a decrease in mRNA levels of both ERs in SAT (*p* < 0.0001) and *ESR2* in VAT (*p* = 0.0001), while weight loss increased *ESR* transcription (*p* < 0.0001). Methylation levels of *ESR1* and *ESR2* promoters were unaffected. However, *ESR1* mRNA in the AT of obese subjects correlated negatively with the expression of hsa-miR-18a-5p (r_s_ = −0.444), hsa-miR-18b-5p (r_s_ = −0.329), hsa-miR-22-3p (r_s_ = −0.413), hsa-miR-100-5p (r_s_ = −0.371), and hsa-miR-143-5p (r_s_ = −0.289), while the expression of *ESR2* in VAT correlated negatively with hsa-miR-576-5p (r_s_ = −0.353) and in SAT with hsa-miR-495-3p (r_s_ = −0.308). In conclusion, obesity-associated downregulation of ER mRNA levels in adipose tissue may result from miRNA interference.

## 1. Introduction

The health risk to an obese individual is determined not only by the amount of adipose tissue but also by its distribution and metabolic activity. In adults, the distribution and metabolism of adipose tissue vary by gender and are regulated by sex steroids, especially estrogens [1]. There is mounting evidence from preclinical studies that estrogens impact adipocytes’ differentiation and secretory activity, adipose tissue lipid storage capacity, and insulin resistance [2]. A decline in estrogens during menopause leads to adverse changes in body composition, negatively affects adipose tissue function, and increases the risk of metabolic complications that can be partially reversed with hormone replacement therapy [3,4]. Notably, however, β-estradiol inhibits glucose utilization in the adipocytes of late postmenopausal women [5].

Estrogens exert their action mostly via the activation of nuclear receptors (ERs), which are present in numerous body cells, including adipocytes. ERs exist in two main forms, namely, α and β, with multiple splice variants that exhibit tissue specificity in expression and function [6]. Classically, ERs act as ligand-dependent transcriptional factors that interact with estrogen response elements in target gene promoters and, in this way, regulate their expression. However, estrogen can also act rapidly via extranuclear and membrane-associated forms of ER (G protein-coupled estrogen receptor (GPER)) that interact with other signaling molecules [7]. Activation of both ERs was found to repress adipogenic differentiation and maturation in mouse bone marrow stromal cells and inhibit adipocyte differentiation, lipid accumulation, and the expression of adipocyte-specific genes in primary human adipocytes [8,9]. In turn, mice of both sexes with a homozygous null mutation for ERα develop obesity from the reduced energy expenditure in the absence of hyperphagia (reviewed in [2]). The role of ERβ in adipose tissue distribution is less studied; however, selective activation of ERβ in adipocytes in vitro induces the expression of genes involved in white adipose tissue browning, while in vivo, it reduces body weight and fat mass in animals on a high-fat diet [10,11]. Both ERα and ERβ were identified in human adipose tissue, but their expressions may differ depending on the adipose tissue depot and donor ethnicity [12,13]. Since ERα- and ERβ-mediated effects may differ and sometimes be opposing, it is postulated that the local proportions of both receptor isoforms are crucial to the tissue-specific response to estrogens [13,14]. However, until now, how obesity and subsequent weight loss influence the expression of both ER isoforms in different adipose tissue depots has not been defined.

Notably, by regulating adipose tissue metabolism and secretory activity, estrogens may modulate the risk of obesity-related complications. In animals, ovariectomy led to insulin resistance and increased susceptibility to the deleterious effects of a high-fat diet, which could be prevented by estrogen supplementation, ensuring the physiological level of this hormone [15]. Decreased levels and/or impaired function of ERα are associated with the increased prevalence of obesity and metabolic syndrome in rodents (reviewed in [2]). In turn, selective ERβ activation leads to lipid mobilization for heat production, which partly corrects the metabolic complications of obesity [11]. In clinical studies, menopause is associated with a constant decline in insulin sensitivity parallel to an increase in serum inflammatory markers and an unfavorable lipid profile [1]. Polymorphisms in genes encoding ERα (*ESR1*) and ERβ (*ESR2*) were associated with the risk of obesity and metabolic syndrome; however, the associations were population-specific [16,17,18,19]. These findings suggested that ERs might constitute targets to combat obesity and its metabolic complications; however, further studies are required to understand the complex role of sex hormones in regulating adipose tissue metabolism.

Epigenetic modifications, such as DNA methylation or microRNA (miRNA) interference, play an essential role in regulating gene expression in adipose tissue, and modulating these processes might be an attractive approach to counteract obesity. Several oncological studies have already demonstrated that DNA methylation and miRNA interference play vital roles in regulating the expression of *ESR1* and *ESR2* [20,21,22].

Despite extensive research on estrogen action in adipose tissue from in vitro and animal studies, the knowledge about obesity-induced changes in estrogen activity in human adipose tissue is limited. Therefore, the main goal of this study was to characterize the ER expression profiles in adipose tissues in normal-weight and obese individuals before and after weight loss and to examine whether epigenetic modifications are involved in regulating genes related to estrogen action in adipose tissue.

## 2. Results

### 2.1. Estrogen Receptors’ Expressions in Adipose Tissues of Obese Individuals before and after Bariatric Surgery and Normal-Weight Subjects

#### 2.1.1. ESR1

In the entire study group, without the gender stratification (Figure 1a), the *ESR1* mRNA expression was lower in the subcutaneous adipose tissue of the obese patients (SAT-O) compared with the SAT of the normal-weight individuals (SAT-N, *p* < 0.0001). This finding was common for the female (*p* < 0.0001) and male (*p* = 0.0001) study participants (Figure 1b and Figure 1c, respectively). Moreover, when the female group was stratified according to their menopausal status, this difference remained significant in both the post- and premenopausal women (*p* = 0.0002, Figure 1d, and *p* < 0.0001, Figure 1e, respectively).

In this study, we had an opportunity to analyze the expressions of genes encoding estrogen receptors in adipose tissues obtained from the patients 18 to 24 months after their bariatric surgery (for the details, please see Section 4). The surgically induced weight loss was associated with a significant increase in the *ESR1* mRNA expression in SAT (SAT-PO) in the whole study group (*p* < 0.0001, Figure 1a), as well as in the male (*p* = 0.002, Figure 1c), all-female (*p* < 0.0001, Figure 1b), and premenopausal female (*p* < 0.0001, Figure 1e) study participants. Since none of the female patients from the weight-loss group had a postmenopausal status, the impact of menopause on this phenomenon could not have been studied. In neither the entire group nor the analyzed subgroups did the *ESR1* mRNA levels in the SAT after weight loss (SAT-PO) differ significantly from that observed in the SAT of normal-weight subjects (SAT-N). When the whole group was analyzed, no differences were found in the *ESR1* mRNA levels between the visceral adipose tissue (VAT) of the obese and normal-weight individuals. However, subgroup analysis revealed that obesity was associated with significant downregulation of *ESR1* mRNA levels in the VAT of the obese premenopausal women compared with the non-obese controls (*p* < 0.0001, Figure 1e). Normal body weight was associated with comparable *ESR1* mRNA levels in VAT and SAT, regardless of gender. In contrast, in the obese subjects, the *ESR1* mRNA expression was higher in VAT compared with SAT in the whole group (*p* = 0.0001, Figure 1a) and females (*p* < 0.0001, Figure 1b), irrespective of the menopausal status (*p* = 0.011 in the postmenopausal group, Figure 1d, and *p* = 0.003 in the premenopausal group, Figure 1e). In the men, the same trend was observed; however, the difference did not reach statistical significance.

#### 2.1.2. ESR2

The *ESR2* mRNA levels were lower in the adipose tissues of the obese individuals (Figure 2a), both in SAT (*p* < 0.0001) and in VAT (*p* = 0.0001), than in the normal-weight controls. The same was found in the women (*p* < 0.0001 for both adipose tissue depots, Figure 2b) and men (*p* = 0.001 for SAT and *p* = 0.0006 in VAT, Figure 2c). Menopausal status did not influence the *ESR2* expression pattern in adipose tissue, as in both the pre- and postmenopausal women, its mRNA levels were lower in obese study participants compared with the normal-weight controls (*p* = 0.007 for both SAT and VAT in postmenopausal women, Figure 2d, and *p* = 0.0004 for SAT and *p* < 0.0001 for VAT in premenopausal women, Figure 2e).

Regardless of sex, weight loss was associated with an increase in the *ESR2* mRNA levels in SAT, with *p* < 0.0001 for the entire group and women (Figure 2a,b,e) and *p* = 0.0022 for the men (Figure 2c). Similar to *ESR1*, the *ESR2* mRNA levels in SAT-PO were not significantly different from the SAT of the normal-weight individuals.

In addition, we found that weight status may influence the the *ESR2* expression proportions between VAT and SAT (Figure 2a). The *ESR2* mRNA levels were lower in VAT compared with SAT in the obese subjects (*p* < 0.0001), while in the normal-weight individuals, no significant difference in the *ESR2* mRNA levels between the two adipose tissue depots was observed. The same situation was seen in the women (*p* < 0.0001, Figure 2b), while in the obese men, no statistically significant differences in the *ESR2* mRNA levels between VAT and SAT were found (Figure 2c). Interestingly, the menopausal status had an influence on the *ESR2* expression in VAT and SAT in the obese women. In the postmenopausal group, as in the men, obesity had no impact on the *ESR2* mRNA concentrations in VAT and SAT (Figure 2d). In contrast, in the premenopausal group (Figure 2e), the *ESR2* mRNA levels in VAT-O were significantly lower than in SAT-O (*p* < 0.0001).

#### 2.1.3. ESR1-to-ESR2 Ratio

Next, we analyzed how obesity and weight loss influenced local proportions of ERα and ERβ mRNA in the different adipose tissue depots.

In the whole group, the *ESR1*/*ESR2* mRNA ratio was higher in the adipose tissues of the obese study participants compared with the normal-weight controls, both in the subcutaneous (*p* = 0.0006) and visceral depots (*p* = 0.02, Figure 3a). However, subsequent gender stratification revealed that these differences were present only in the females (*p* = 0.0003 and *p* = 0.04, respectively, Figure 3b), while in men, no significant differences in the *ESR1*/*ESR2* mRNA ratio between the obese and normal-weight patients were observed (Figure 3c). When the female group was divided according to their menopausal status, in the adipose tissues of obese premenopausal women, the *ESR1*/*ESR2* mRNA ratio was higher in both SAT and VAT (*p* = 0.03 and *p* = 0.02, respectively, Figure 3e) compared with the normal-weight controls, while in the postmenopausal women, the ratio was higher in SAT only (*p* = 0.014, Figure 3d).

Importantly, in all the studied groups and subgroups, the proportion between *ESR1* and *ESR2* mRNA levels was higher in the subcutaneous adipose tissue of the obese-after-weight-loss individuals compared with the obese-before-weight-loss patients (Figure 3a–e). Moreover, the *ESR1*/*ESR2* mRNA ratio was higher in the SAT of the obese-after-weight-loss patients compared with the SAT of the normal-weight controls in the whole investigated group (*p* < 0.0001, Figure 3a), in all the females (*p* < 0.0001, Figure 3b), and in the premenopausal women (*p* < 0.0001, Figure 3e).

Finally, we found that in the obese study participants, the proportion between the *ESR1* and *ESR2* mRNA levels was higher in the visceral than in the subcutaneous adipose tissue depots; however, in the postmenopausal women, the difference did not reach statistical significance.

### 2.2. Estrogen Receptors’ Expressions in Adipose Tissues of Obese Individuals Stratified by the Presence of Obesity-Related Co-Morbidities

In order to investigate the association between obesity-related changes in the expression of *ESR1* and *ESR2* in adipose tissue and the development of metabolic complications of obesity, we compared their mRNA concentrations in tissues of the obese patients stratified by the presence of the chief components of metabolic syndrome: prediabetes/type 2 diabetes, hyperlipidemia, and hypertension.

When the entire group of obese study participants was divided into subgroups according to their glucose tolerance statuses (Figure 4a), the *ESR1* mRNA levels were found to be significantly lower in the SAT of the obese subjects with type 2 diabetes and prediabetes (DM) compared with those with normal glucose levels (NDM, *p* = 0.016). However, gender stratification revealed that this was observed in the women only (*p* = 0.019, Figure 4b). Moreover, further stratification of the female patients by menopausal status revealed that the association between the *ESR1* mRNA levels in SAT and diabetes was limited to the premenopausal women only (*p* = 0.0012, Figure 4e), while in the postmenopausal group, as in the men, no difference between the diabetic and nondiabetic patients was observed (Figure 4d). No statistically significant differences were found in the expression of *ESR1* in VAT between the diabetic and nondiabetic patients in any of the studied groups and subgroups (Figure 4a–e). There was also no association between the concentrations of *ESR2* mRNA in VAT and SAT and the value of the *ESR1*/*ESR2* ratio and the incidence of diabetes or prediabetes in the obese patients.

Contrary to participants with diabetes, stratification of the obese study participants according to the presence of hyperlipidemia or hypertension revealed no significant differences in the *ESR1* and *ESR2* mRNA levels and the *ESR1*/*ESR2* ratio between the affected and non-affected individuals, irrespective of gender and menopausal stratification. Moreover, no significant differences between the *ESR1* and *ESR2* mRNA levels and the *ESR1*/*ESR2* ratio were found in the adipose tissues of the obese individuals meeting the diagnostic criteria of metabolic syndrome and those who were metabolically healthy.

### 2.3. DNA Methylation in the Regulation of Estrogen Receptors’ Expressions in Adipose Tissue

In order to determine whether epigenetic modifications play a role in regulating obesity-related changes in *ESR1* and *ESR2* mRNA levels in adipose tissues, we investigated the methylation status of the regulatory regions of these two genes.

#### 2.3.1. ESR1

The 5’ regulatory region of *ESR1* consists of seven independent promoters [23]. A 3 kb fragment of *ESR1* located from 2.5 kb upstream to 0.5 kb downstream of the main transcription start site (TSS transcript ENST00000206249.8 release of the ENSEMBL database) was analyzed using CpG Islands prediction tools. In silico analysis identified a 649 bp fragment within the *ESR1* gene, located between −34 bp and +615 bp relative to this TSS, with 62 potential methylation sites, corresponding to the promoter α region [24]. Owing to the methodological limitations of the methylation analysis, this CpG island was divided into two parts, which were separately analyzed (Appendix A); however, the mean percentage of the methylated CG pairs in both fragments was similar (*p* > 0.05) irrespective of (i) weight status, (ii) presence of the metabolic complications of obesity, and (iii) adipose tissue origin (visceral or subcutaneous) (Appendix A). Notably, there was no correlation between the methylation and *ESR1* mRNA levels, suggesting that methylation does not play a role in obesity-related *ESR1* expression changes in adipose tissue.

#### 2.3.2. ESR2

*ESR2* has two promoters—distal (P0K) and central (P0N)—while some authors also distinguish a third, namely, proximal promoter (PE1). The 44,481 bp long intronic sequence separates P0K (199 bp) from P0N (328 bp), which, in turn, is separated by a 10,957 bp intron from the first coding exon [25]. In silico analysis of the *ESR2* fragment located 60 kb upstream of the TSS (transcript ENST00000341099.6 release of the ENSEMBL database) revealed two CpG islands: 661 bp in the P0K region (−56,044 to −55,433 bp relative to the TSS), including 49 potential methylation sites, and 641 bp in the P0N region (−11,532 to −10,891 bp relative to the TSS) with 41 potential methylation sites. We analyzed both CpG islands, and each island was additionally divided into two parts for methodological reasons (Appendix A). However, the methylation status of the analyzed regions did not correlate with the *ESR2* mRNA levels, Moreover, no significant differences in the methylation statuses were found between the DNA samples obtained from the VAT and SAT of (i) obese and normal-weight study participants and (ii) metabolically healthy obese subjects and those with the metabolic complications of obesity (Appendix A).

### 2.4. The Role of miRNA Interference in Regulating Estrogen Receptors’ Expressions in Adipose Tissue

Based on the results of the miRNome analysis with the use of the next-generation sequencing method and subsequent bioinformatic analysis with the MirWalk and MirTarBase programs [26], we hypothesized that genes encoding estrogen receptors are possible targets of miRNAs that are differentially expressed in the investigated tissues. Upon the in silico analysis, as well as based on the previously published data reporting the results of functional studies performed on human biological material, we selected hsa-miR-18a-5p, hsa-miR-18b-5p, hsa-miR-22-3p, hsa-miR-100-5p, hsa-miR-142-3p, and hsa-miR-143-5p for the analysis of *ESR1*, and hsa-miR-146b-3p, hsa-miR-20b-5p, hsa-miR-335-3p, hsa-miR-495-3p, and hsa-miR-576-5p for the analysis of *ESR2* [21,22,27,28,29,30]. We previously found that all these miRNAs have significantly different expression levels between adipose tissues obtained from obese and normal-weight individuals [26].

#### 2.4.1. ESR1

In the entire study group, significant negative correlations between the *ESR1* mRNA levels in SAT-O and hsa-miR-18a-5p (*p* = 0.0003, r_s_ = −0.444, Figure 5a), hsa-miR-18b-5p (*p* = 0.009, r_s_ = −0.329, Figure 5b), hsa-miR-22-3p (*p* = 0.004, r_s_ = −0.413, Figure 5c), hsa-miR-100-5p (*p* = 0.003, r_s_ = −0.371, Figure 5d), and hsa-miR-143-5p (*p* = 0.02, r_s_ = −0.289, Figure 5e) were found.

A negative correlation between *ESR1* mRNA levels and hsa-miR-18a-5p was observed in both the female and male obese study participants (*p* = 0.002, r_s_ = −0.456 and *p* = 0.009, r_s_ = −0.794, respectively, Appendix A). Similarly, in SAT-O, the hsa-miR-18b-5p levels correlated negatively with *ESR1* mRNA in the women (*p* = 0.047, r_s_ = −0.308, Appendix A) and men (*p* = 0.044, r_s_ = −0.661, Appendix A). Otherwise, correlations between hsa-miR-22-3p (*p* = 0.037, r_s_ = −0.371, Appendix A) and hsa-miR-100-5p (*p* = 0.0036, r_s_ = −0.439, Appendix A) were found only in the women. Menopausal status also had an impact on the observed phenomena: in the premenopausal women, significant negative correlations were observed in SAT-O between hsa-miR-18a-5p (*p* = 0.002, r_s_ = −0.456, Appendix A), hsa-miR-18b-5p (*p* = 0.04, r_s_ = −0.308, Appendix A), hsa-miR-22-3p (*p* = 0.037, r_s_ = −0.371, Appendix A), hsa-miR-100-5p (*p* = 0.004, r_s_ = −0.439, Appendix A), and hsa-miR-143-5p (*p* = 0.028, r_s_ = −0.352, Appendix A), while in the postmenopausal women, no significant correlations between the abovementioned miRNAs and *ESR1* mRNA were found.

Subsequently, we investigated the possible role of miRNA interference in regulating *ESR1* mRNA levels in subcutaneous adipose tissues derived from obese individuals stratified by the presence of diabetes and prediabetes. However, no significant correlations were observed between the investigated miRNAs and *ESR1* mRNA levels in the SAT of the diabetic and nondiabetic study participants. Furthermore, no significant correlations between the expression of *ESR1* and the abovementioned miRNA were observed in the adipose tissue samples originating from the normal-weight individuals, as well as in the SAT of the postbariatric patients.

#### 2.4.2. ESR2

Next, we searched for the correlations between the *ESR2* mRNA levels and hsa-miR-146b-3p, hsa-miR-20b-5p, hsa-miR-335-3p, hsa-miR-495-3p, and hsa-miR-576-5p concentrations in the adipose tissues of the study participants. When the entire group of obese individuals was analyzed, we found significant negative correlations between the *ESR2* mRNA expression and hsa-miR-576-5p (*p* = 0.01, r_s_ = −0.353, Figure 6a) in VAT and hsa-miR-495-3p (*p* = 0.01, r_s_ = −0.308, Figure 6b) in SAT.

After gender stratification, a significant correlation between the *ESR2* mRNA levels and hsa-miR-576-5p in VAT-O was observed in the women (*p* = 0.007, r_s_ = −0.411, Appendix A). Subsequent subgroup analysis revealed that this correlation was significant in the premenopausal study participants only (*p* = 0.002, r_s_ = −0.511, Appendix A). The subgroup analysis did not identify significant correlations between the *ESR2* mRNA expression and hsa-miR-495-3p in SAT-O; however, a trend was observed in the female subgroup (*p* = 0.07, r_s_ = −0.262, Appendix A). As in the case of *ESR1*, the expression of *ESR2* did not correlate with the expression of the selected miRNAs in the tissues of the healthy individuals and those after bariatric surgery.

## 3. Discussion

This study aimed to assess the impact of obesity and weight loss on the transcriptional activity of genes encoding estrogen receptors α and β (*ESR1* and *ESR2*, respectively) in adipose tissues and to verify whether epigenetic mechanisms, namely, DNA methylation and miRNA interference, might mediate this phenomenon. We found that obesity was associated with a significant decrease in *ESR1* and *ESR2* mRNA levels in adipose tissue; however, the effect was depot- and gender-specific. Our findings suggested that regardless of gender, weight loss could restore *ESR1* and *ESR2* mRNA concentrations to those observed in normal-weight individuals. The obesity-related changes in *ESR1* and *ESR2* expression in adipose tissues did not correlate with the methylation status of the regulatory regions in these two genes. However, we found significant negative correlations between the *ESR1* and *ESR2* mRNA levels and the concentrations of several miRNAs known to target complementary sequences in their 3′ untranslated regions (UTRs).

Our finding regarding an inverse association between the *ESR1* expression in adipose tissue and obesity was consistent with previous human studies [31,32]. Analyzing SAT samples from 16 non-obese and 17 obese premenopausal women, Nilson et al. found, similar to our dataset, that obesity is associated with decreased *ESR1* mRNA levels. Moreover, they also observed a trend toward an increase in *ESR1* expression in SAT after weight reduction in seven out of nine investigated patients [31]. Recently, Zhou et al. reanalyzed data from three trials evaluating the correlation between adiposity and *ESR1* mRNA levels in SAT biopsies of 766 women (median age 62, ~75% postmenopausal, participants of the TwinsUK study); 24 male participants of the Skeletal Muscles, Myokines, and Glucose Metabolism (MyoGlu) study; and 770 men enrolled in the Metabolic Syndrome in Men (METSIM) study [32,33,34,35]. Like ours, their results unequivocally showed that *ESR1* expression in SAT was inversely correlated with adipose tissue content. In contrast, data on the influence of weight status on *ESR2* mRNA levels in adipose tissue are scarce; however, our study suggested that the obesity and weight-loss-related changes in *ESR2* expression in SAT mimic those observed in the case of *ESR1*.

Our study also investigated the impact of obesity on ER expression in the visceral adipose tissue depot. In the entire group and the studied subgroups, *ESR2* mRNA levels in the VAT of the obese study participants were significantly lower than those of the normal-weight individuals. Interestingly, in contrast to *ESR1* expression in SAT, obesity did not seem to induce significant changes in *ESR1* expression in VAT. This phenomenon was present in the whole study group and for the women and men. However, stratification by the menopausal status revealed that the VAT of the obese premenopausal women was characterized by a significant decline in the *ESR1* mRNA concentrations compared with the VAT of the normal-weight age-matched controls. To our knowledge, none of the previous studies analyzed the combined impact of menopause and weight changes on the ER expression in VAT. Nevertheless, our finding is in accordance with the results of animal studies suggesting that surgically induced menopause may lead to an increase in ER receptor levels in adipose tissue that is interpreted as a compensatory mechanism in response to declines in estrogen levels [36]. Moreover, studies performed on human SAT confirmed the hypothesis of the differential influence of menopause on ER status in adipose tissue. For instance, Park et al. found that *ESR1* expression in the SAT of 22 postmenopausal women was significantly lower than in 23 premenopausal study participants, while the expression of *ESR2* did not change with age [14].

In addition, while there was no significant difference between ER mRNA expression between the two adipose tissue depots in the normal-weight subjects, the *ESR1* mRNA levels in VAT in the obese study participants were higher than the *ESR2* mRNA concentrations, while the opposite trend was found in SAT. Both physiological (e.g., aging, menopause) and pathological conditions (including overweight and obesity) may disturb the ERα/ERβ ratio in adipose tissue and, in this way, modify the profile of estrogen actions [37]. In our study, the abdominal SAT of normal-weight individuals was characterized by a lower *ESR1*/*ESR2* mRNA ratio than the tissues obtained from the obese patients, regardless of gender and menopausal status. The obesity-related increase in the *ESR1*/*ESR2* mRNA ratio was also present in VAT; however, the difference did not reach statistical significance in the men or postmenopausal women. This finding agreed with previous reports, where the value of the ERα/ERβ protein ratio correlated positively with BMI and waist circumference in women [37]. Interestingly, in the obese patients after weight loss, the *ESR1*/*ESR2* mRNA ratio in SAT and VAT remained elevated, suggesting that obesity may leave a permanent mark on adipose tissue function.

Since preclinical studies have suggested that ERs act as negative regulators of adipogenesis and lipogenesis, it can be presumed that their diminished levels in the adipose tissue of obese individuals may, in the vicious circle mechanism, predispose them to additional weight gain [2,8,9]. One may wonder about the primary phenomenon: whether excess adiposity causes a decrease in *ESR1* and *ESR2* expression or whether their decreased levels predispose humans to obesity. On the one hand, the finding that weight loss partially restores ER mRNA levels in SAT to those observed in normal-weight individuals suggested that a decline in the expression of these genes is secondary to weight gain. On the other hand, as the menopausal status may impact ER mRNA levels in adipose tissue, a menopause-related decrease in ER expression may make postmenopausal women more prone to weight gain and metabolic disturbances.

Epidemiological studies showed that women after menopause and age-matched men are characterized by increased insulin resistance compared with premenopausal women, and the disparity is attributed to lower estrogen levels [38]. These findings were consistent with the results of the preclinical experiments showing that estrogens, via activation of *ESR1*, can regulate insulin signaling and mitochondrial metabolism, contributing to the improvement of insulin sensitivity in adipose tissue [32,39]. In addition, Orozco et al., when analyzing the abovementioned MyoGlu cohort, reported a significant decrease in *ESR1* mRNA in SAT samples from dysglycemic men compared with normoglycemic controls [34]. The same was observed in our study in the entire group and in females, while in the males, the difference did not reach statistical significance, possibly because of a low number of analyzed samples. Interestingly, stratification according to menopausal status revealed that the association between lower *ESR1* expression in SAT and prevalence of diabetes was observed in the premenopausal women only. This finding suggested that an obesity-associated decrease in *ESR1* mRNA concentration in adipose tissue may eliminate estrogen’s protective, anti-diabetic effect in premenopausal women.

The beneficial effect of estrogens on adipose tissue metabolism is not limited to the regulation of insulin sensitivity. Estradiol was found to control lipoprotein lipase activity in a dose-dependent manner in vitro and in vivo [40]. At the same time, in clinical studies, its transdermal administration decreased the expression of genes encoding critical lipogenic enzymes (stearoyl-CoA desaturase, fatty acid synthase, acetyl-coenzyme A carboxylase alpha, fatty acid desaturase 1) in human SAT that correlated with a decrease in plasma triglyceride levels [41]. However, our work did not find any difference in ER mRNA levels in adipose tissues originating from the obese individuals stratified by hyperlipidemia. Estrogens were also found to exert anti-inflammatory effects in adipocytes in vitro and in vivo, and in this way, to prevent obesity-associated adipose tissue dysfunction and subsequent metabolic complications of obesity (reviewed in [42]). Therefore, we analyzed whether the adipose tissues originating from the obese study participants meeting the diagnostic criteria of the metabolic syndrome differed in ER levels from those who were metabolically healthy. However, there was no association between the *ESR1* and *ESR2* mRNA levels and the prevalence of metabolic syndrome in the studied cohort. Similarly, we did not find significant differences in the ER mRNA concentrations between the tissues obtained from the patients stratified by the presence of hypertension. Nevertheless, one should remember that the mechanisms underlying the development of obesity’s metabolic complications are complex, and the determination of the estrogen concentration in the examined tissues would be decisive for explaining a cause-and-effect relationship for these findings.

Even though the association between the decline in *ESR1* and *ESR2* expression in adipose tissue and the incidence of obesity has been postulated for some time, little is still known about the molecular mechanisms leading to these phenomena [14,31,32,33,34,35]. Given the role of epigenetic mechanisms in regulating the expression of genes encoding ERs in estrogen-dependent tumors, we investigated whether DNA methylation and miRNA interference participated in the observed obesity-induced decrease in the *ESR1* and *ESR2* mRNA levels in adipose tissue.

We analyzed CpG islands located in the regions corresponding to the *ESR1* promoter α, as well as to the distal (P0K) and central (P0N) *ESR2* promoters [24,25]. The aberrations in methylation status within these regions were previously associated with changes in *ESR1* and *ESR2* transcriptional activity in estrogen-dependent cancers and endometriosis [20,43,44]. However, we did not observe any significant correlations between the percentage of the methylated CG pairs in the analyzed DNA fragments and the ER mRNA levels in the investigated tissues. The methylation status of the analyzed regions did not differ significantly between DNA samples derived from the VAT and SAT of the individuals stratified by weight status or presence of the metabolic complications of obesity. There are several possible explanations for this finding. First, our results may reflect the fact that methylation does not play an essential role in obesity-related changes in ER expression in adipose tissue. However, it is also possible that the analyzed fragments of the *ESR1* and *ESR2* promoters were not crucial for regulating their expression or that the number of the DNA samples included in the methylation analysis was not sufficient to detect significant differences between the analyzed subgroups. Moreover, since DNA isolation was performed using adipose tissue homogenates in our study, cell-type heterogeneity could have impacted the obtained results. Apart from adipocytes, adipose tissue contains several other cell types, including, among others, stromal and immune cells (e.g., macrophages), as well as cells of blood vessels. These cells vary substantially in terms of their DNA methylation profiles. Therefore, further studies, including single-cell analyses, are required to verify the role of DNA methylation in regulating obesity and weight-loss-associated changes in *ESR1* and *ESR2* transcriptional activity in adipose tissue [45].

Studies in recent years have highlighted the role of miRNAs as important regulators of gene expression in adipose tissue. It was also reported that the expression of miRNAs involved in the regulation of adipogenesis and adipose tissue metabolism differs in the adipose tissues of obese patients and that of individuals with a normal body weight (reviewed in [46]). Moreover, several miRNAs were shown to regulate *ESR1* and *ESR2* expression in estrogen-dependent tissues [47,48]. Therefore, we verified whether the observed obesity and weight-loss-associated differences in the concentrations of the ER mRNA levels in adipose tissues resulted from miRNA interference. We selected miRNAs for this analysis that met three primary criteria: (i) a predicted in silico interaction with *ESR1* or *ESR2* mRNA, (ii) a suggested role in the regulation of *ESR1* or *ESR2* expression in vivo, and (iii) an altered expression in adipose tissue in the course of obesity or weight loss.

In five out of six cases of selected miRNAs that potentially interact with *ESR1* 3’UTR, we found significant negative correlations with the *ESR1* mRNA levels in the adipose tissues of the obese study participants. Notably, hsa-miR-18a and -18b are known to regulate *ESR1* expression in cancer tissues [21,49]. A high expression of these two miRNAs is associated with increased proliferation and a worse prognosis in ER-negative breast cancers [21]. In our study, high levels of these miRNAs correlated negatively with *ESR1* mRNA concentrations in the SAT of obese patients of both genders. However, stratification according to the menopausal status revealed significant correlations of *ESR1* mRNA level with hsa-miR-18a and hsa-miR-18b concentrations in the SAT of premenopausal women only, suggesting that miRNAs may partially mediate menopause-related changes in adipose tissue function. To date, little is known about hsa-miR-18a and hsa-miR-18b in adipose tissue development. However, hsa-miR-18b, through regulation of the crucial macrophage transcription factors, plays a significant role in macrophage lineage development and polarization toward a proinflammatory M1 lineage [50]. Therefore, overexpression of this miRNA in obesity may lead to the increased production of proinflammatory cytokines (e.g., interleukins 1β and 6) and aggravate adipose tissue dysfunction, despite proper estrogen levels. hsa-miR-22-3p is an example of another miRNA that is involved in the regulation of the proinflammatory response, whose expression correlated negatively with *ESR1* mRNA levels in the SAT of obese participants in our study. Since deletion of this miRNA attenuates the accumulation of adipose tissue by reducing white adipocyte differentiation in animal models, its increased concentrations in human SAT during obesity may not only neutralize the beneficial effects of *ESR1* activation but also promote excess fat accumulation [27,51]. hsa-miR-100-5p and hsa-miR-143-5p are the two molecules that regulate the response to therapy in ER-positive breast cancers [52,53]. The overexpression of these two miRNAs was reported in the adipose tissue of obese individuals [26,54,55]. However, preclinical studies suggested their different roles in the development of obesity-associated complications: while the overexpression of miR-100 protects against liver steatosis, hypertriglyceridemia, and the development of metabolic syndrome in mice, the obesity-related increase in miR-143 levels predisposes them to insulin resistance [56,57]. In our study, we found negative correlations between the expressions of these two miRNAs and *ESR1* mRNA levels in the SAT of the obese study participants. However, as in the case of hsa-miR-22-3p, the correlations between the hsa-miR-100-5p and hsa-miR-143-5p levels and *ESR1* mRNA concentrations were dependent on gender and menopausal status, as we detected them in the female obese study participants and in the premenopausal subgroup, but not in the adipose tissues derived from men or the postmenopausal women.

The roles of miRNAs in regulating *ESR2* transcriptional activity have been much less studied. What distinguished the obesity-related changes in *ESR2* expression compared with *ESR1* in our study was that the decrease in *ESR2* gene expression referred to visceral and subcutaneous adipose tissue depots. Subsequently, we found that the interference from two distinct miRNAs could be co-responsible for this phenomenon: in VAT-O, we observed a significant negative correlation between *ESR2* mRNA levels and hsa-miR-576-5p, while SAT-O was significantly negatively correlated with hsa-miR-495-3p.

hsa-miR-576-5p has an established position as a regulator of carcinogenesis and inflammation [58,59]. Its high serum levels were detected in obese adolescents and positively correlated with higher concentrations of proinflammatory cytokines [60]. Interestingly, both hsa-miR-576-5p and ERβ act as modulators of the Wnt/β-catenin signaling pathway, regulating lipogenesis and inflammatory responses in adipose tissue [30,61]. However, while ERβ acts as a positive regulator of Wnt/β-catenin signaling, hsa-miR-576-5p inhibits this pathway. Therefore, hsa-miR-576-5p-dependent downregulation of *ESR2* expression in VAT can promote metabolic inflammation. In turn, in animal models with diet-induced obesity and type 2 diabetes, miR-495 induced the transformation of adipose tissue macrophages into M1 proinflammatory type and enhanced insulin resistance [62]. One can suppose that the obesity-induced upregulation of hsa-miR-495-3p in SAT may neutralize the beneficial influence of estrogen (via interference with *ESR2* mRNA) and promote adipose tissue inflammation and the development of metabolic complications of obesity. Importantly, gender stratification revealed a possible role of estrogens in regulating the miRNA–*ESR2* interference: the significant negative correlations were observed in females only, while in the case of hsa-miR-576-5p, they were also observed in the premenopausal subgroup. An interesting aspect of epigenetic regulation of ERs in adipose tissue is that they are not only subjects of epigenetic mechanisms but also, by regulating several miRNA expressions and gene methylation statuses, their essential mediators, depending on the local estrogen status [63]. This finding can partially explain the abovementioned differences in the observed correlations between the genders and women stratified by menopausal status.

Notably, all the correlations mentioned above between the selected miRNAs and ER’s mRNA levels require verification in functional studies in human adipocytes since a particular miRNA playing a role in regulating gene transcriptional activity in one tissue does not have to translate into another.

This study had several limitations. First, the *ESR1* and *ESR2* expression analysis was performed at an mRNA level only. Even though our results are consistent with those from the previous studies in human adipose tissues, the observed changes in ER transcriptional activity require verification at the protein level. Second, the number of men included in the subgroup analysis was limited; therefore the results obtained in this group should be treated with caution. Next, the study had a cross-sectional character because the patients from the postoperative (PO) group were independently recruited. Moreover, no women in the PO group had a postmenopausal status. Finally, the study design made the obtained data mainly descriptive and functional studies on human adipocyte cell lines are required to evaluate the biological value of the findings regarding miRNA–mRNA interference and to verify the role of DNA methylation in the regulation of ER gene expression in adipose tissue during obesity. However, this work also had several strengths. To the best of our knowledge, it is the first study that reported how obesity and weight loss impact ER balance in adipose tissue and explored the molecular mechanisms underlying these phenomena. Moreover, it raised the possibility of analyzing how gender and menopausal status influence *ESR1* and *ESR2* expression in adipose tissue in a weight-dependent manner. Ultimately, it offered insight into how body mass, gender, and menopause-related decline in estrogen levels modulate ER transcriptional activity, not only in subcutaneous but also in visceral adipose tissue depots.

## 4. Materials and Methods

### 4.1. Studied Groups and Tissues

Study participants were recruited from the Department of General and Transplantation Surgery; the Medical University of Warsaw; and the Department of General, Oncological and Bariatric Surgery, Medical Centre of Postgraduate Education Warsaw, as described previously [26,64]. The group of obese patients consisted of 78 individuals with a body mass index (BMI) > 40  kg/m^2^: 65 females (47 premenopausal and 18 postmenopausal) and 13 males, whose basic clinical and biochemical characteristics are summarized in Table 1. Postmenopausal status was defined as 12 consecutive months without menstruation in women after 45 years.

In the whole group, prediabetes (impaired glucose tolerance or impaired fasting glucose) or type 2 diabetes was diagnosed in 27 (34.6%) obese study participants (23 females and 4 males). In turn, 43 (55.1%) obese patients (31 females and 12 males) were diagnosed with hypertension and 46 (59.0%) (34 females and 12 males) with hyperlipidemia. Based on the International Diabetes Federation criteria for Europeans [65], metabolic syndrome was diagnosed in 41 (52.5%) (29 females and 12 males) obese study participants, while 13 participants (females only) had no medical history of cardio-metabolic complications of obesity.

All study participants underwent surgical treatment for obesity (sleeve gastrectomy or mini-gastric bypass), and during the procedures, samples of VAT (N = 78) and abdominal SAT (N = 78) were collected. Nineteen additional samples of abdominal SAT were collected from formerly obese subjects (15 females and 4 males) 18 to 24 months after surgery-induced weight loss (PO, BMI 24.3–29.5 kg/m^2^). Six patients from the post-bariatric group suffered from residual hypertension, three from prediabetes, and six from hyperlipidemia; however, none met the criteria for metabolic syndrome. Since post-bariatric surgery is not associated with opening the abdominal cavity, VAT samples from the PO subjects were unavailable.

In total, 55 control tissues were collected from 31 normal-weight individuals (N, BMI 20–24.9 kg/m^2^): 19 females (9 premenopausal and 10 postmenopausal) and 12 males undergoing elective cholecystectomy (24 samples of VAT and 24 samples of SAT) or operated on for an inguinal hernia (7 samples of SAT). They had no medical history of chronic diseases, including the components of the metabolic syndrome: hypertension, diabetes, and hyperlipidemia. Although their adipose tissue content and waist circumference were not available, these subjects were considered to be metabolically healthy based on their medical history, BMI values, and blood test results.

The project was approved by the Bioethics Committees (decision nos. KB 147/2009—obtained on 28th July 2009, KB 91/A/2010—obtained on 19th July 2010, and KB 117/A/2011—obtained on 14th November 2011) of the Medical University of Warsaw and the Medical Centre of Postgraduate Education in Warsaw, and all participants signed informed consent for participation in this study.

### 4.2. Nucleic Acid Isolation, Reverse Transcription, and Real-Time PCR

Previously described methods [66,67] were used for nucleic acid isolation and reverse transcription of total and miRNA to complementary DNA (cDNA). Real-time PCR was performed in a LightCycler 480 Instrument II (Roche, Mannheim, Germany) for mRNA analysis with a LightCycler 480 Sybr Green I Master Kit (Roche, Mannheim, Germany) [66]. The miRNA analysis was performed with a miRCURY™ LNA™ Universal RT microRNA PCR system (Qiagen, Hilden, Germany—previously Exiqon, Vedbaek, Denmark), which is a microRNA-specific system designed for detecting miRNA via quantitative real-time PCR using SYBR Green as previously described [26,64,65]. Briefly, 100 ng of total RNA was used to synthesize cDNA. Subsequently, the cDNA was diluted in RNAse-free dH_2_O(Invitrogen, Carlsbad, CA, USA). Next, 4 μL of the diluted cDNA (corresponding to 1 ng of initial total RNA) was used for the real-time reaction that contained 5 μL of the SYBR Green Master Mix and 1 μL of the primer mix. The PCR conditions were as follows: denaturation at 95 °C for 10 min, 45 cycles of 95 °C for 10 s and 61 °C for 60 s, and then one melting curve cycle (to verify the specificity of the PCR products). All measurements were performed in triplicate. The results, which were normalized against the expression of the β-actin gene (*ACTB*, for mRNA) and hsa-miR-103a-3p (the recommended control miRNA for the adipose tissue [68]), are presented in arbitrary units (AUs) as mean/median mRNA/miRNA levels. The real-time PCR conditions are summarized in Appendix A. The *ESR1*/*ESR2* ratio was defined as the quotient of the normalized expression of *ESR1* to *ESR2* in each of the examined tissues.

### 4.3. Methylation Analysis

Methylation analysis was preceded by an in silico search of the regulatory regions of *ESR1* and *ESR2* with the CpG Plot (http://www.ebi.ac.uk, accessed on 23 July 2021) and CpG Islands Searcher (http://cpgislands.usc.edu, accessed on 23 July 2021) tools. Sixty DNA samples originating from the VAT and SAT of 20 obese patients (5 metabolically healthy and 15 with metabolic complications of obesity: hypertension, diabetes, hyperlipidemia) and 10 normal-weight individuals were selected for methylation analysis. Analysis was performed using the methylation-sensitive digestion/real-time PCR method with the OneStep qMethyl Kit (Zymo Research, Irvine, CA, USA) according to the manufacturer’s protocol, as described previously [67,69]. Primers used in the methylation analysis are presented in Appendix A.

### 4.4. Statistical Analysis

The normality of the distribution and homogeneity of the variance of the studied parameters were checked with the Shapiro–Wilk and Levene’s tests, respectively. The differences in mRNA, miRNA, and methylation of *ESR1* and *ESR2* CpG island levels were calculated using the Student’s *t*/Mann–Whitney U tests. Correlations between the quantitative values were analyzed with the Spearman correlation test. All statistical analyses were performed with the Statistica software package v.10 (StatSoft, Tulsa, OK, USA) and GraphPad Prism software v.7 (GraphPad Software, San Diego, CA, USA).

## 5. Conclusions

In the present study, we found that obesity and weight loss impacted the transcriptional activity of genes encoding estrogen receptors in adipose tissue. Excess adiposity was associated with a significant decrease in the *ESR1* and *ESR2* mRNA levels in adipose tissue in a depot- and gender-specific manner, and this decrease may predispose patients to the development of diabetes. Even though weight loss restored the *ESR1* and *ESR2* mRNA concentrations to those observed in normal-weight individuals of both sexes, the *ESR1* to *ESR2* ratio remained elevated in the tissues of post-bariatric patients, suggesting that former obesity had a permanent impact on their adipose tissue physiology. Our results also suggested that the abovementioned phenomena may result from the interference between the ESR1 and *ESR2* mRNA and miRNAs rather than from the hypermethylation of the regulatory regions within these genes. However, this finding requires further confirmation in functional studies.

## Figures and Tables

**Figure 1 ijms-23-05989-f001:**
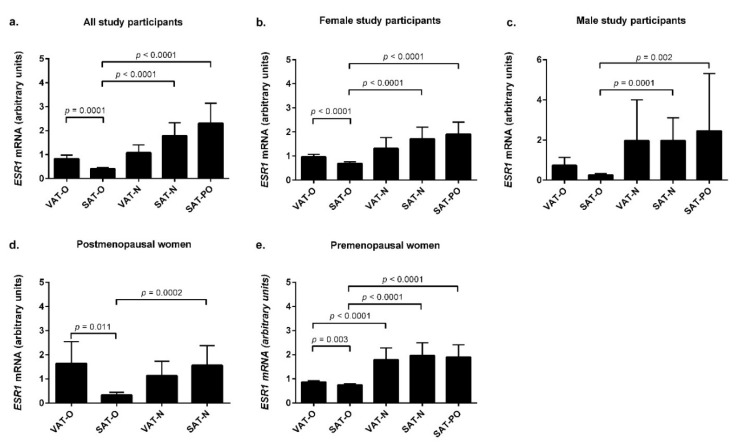
*ESR1* mRNA levels in the visceral (VAT) and subcutaneous (SAT) adipose tissues of the obese individuals before (O) and after surgically induced weight loss (PO) and in the normal-weight subjects (N). Results are presented as the median with the interquartile range for the whole group (**a**), women (**b**), men (**c**), postmenopausal women (**d**), and premenopausal women (**e**).

**Figure 2 ijms-23-05989-f002:**
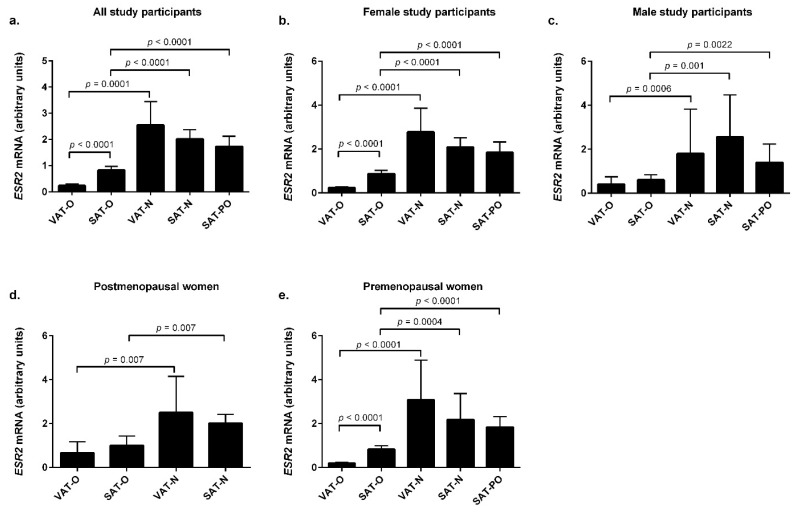
*ESR2* mRNA levels in the visceral (VAT) and subcutaneous (SAT) adipose tissues of obese individuals before (O) and after surgically induced weight loss (PO) and in normal-weight subjects (N). Results are presented as the median with the interquartile range for the whole group (**a**), women (**b**), men (**c**), postmenopausal women (**d**), and premenopausal women (**e**).

**Figure 3 ijms-23-05989-f003:**
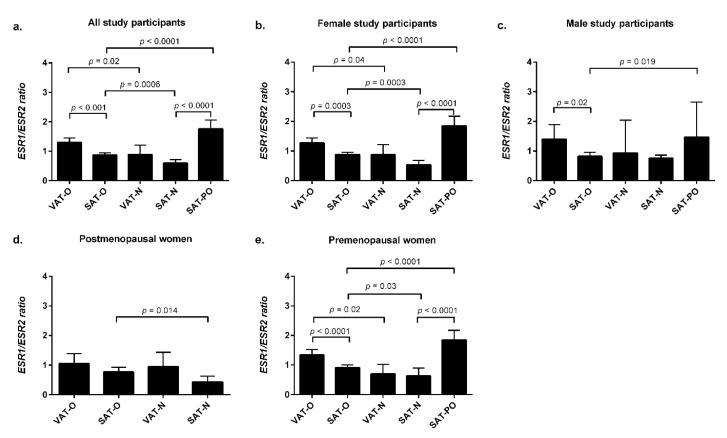
The *ESR1*/*ESR2* mRNA ratio in the visceral (VAT) and subcutaneous (SAT) adipose tissues of obese individuals before (O) and after surgically induced weight loss (PO) and in normal-weight subjects (N). Results are presented as the median with the interquartile range for the whole group (**a**), women (**b**), men (**c**), postmenopausal women (**d**), and premenopausal women (**e**).

**Figure 4 ijms-23-05989-f004:**
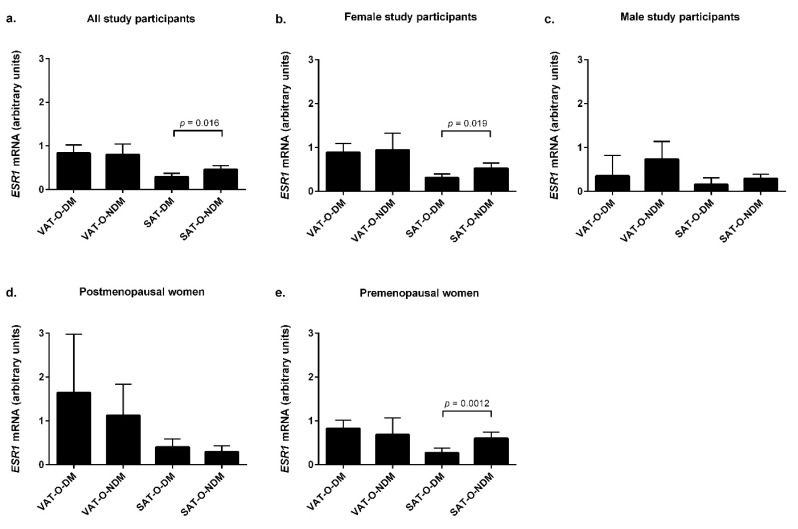
*ESR1* mRNA levels in the visceral (VAT) and subcutaneous (SAT) adipose tissues of obese individuals (O) stratified by the presence (DM) or absence (NDM) of type 2 diabetes. Results are presented as the median with the interquartile range for the whole group (**a**), women (**b**), men (**c**), postmenopausal women (**d**), and premenopausal women (**e**).

**Figure 5 ijms-23-05989-f005:**
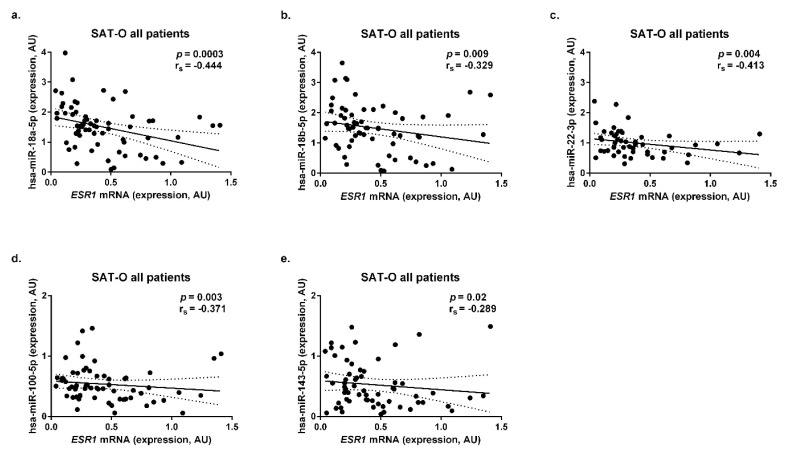
Correlations between *ESR1* mRNA concentrations and hsa-miR-18a-5p (**a**), hsa-miR-18b-5p (**b**), hsa-miR-22-3p (**c**), hsa-miR-100-5p (**d**), and hsa-miR-143-5p (**e**) levels in the subcutaneous adipose tissue (SAT) of all obese (O) study participants. AU—arbitrary units.

**Figure 6 ijms-23-05989-f006:**
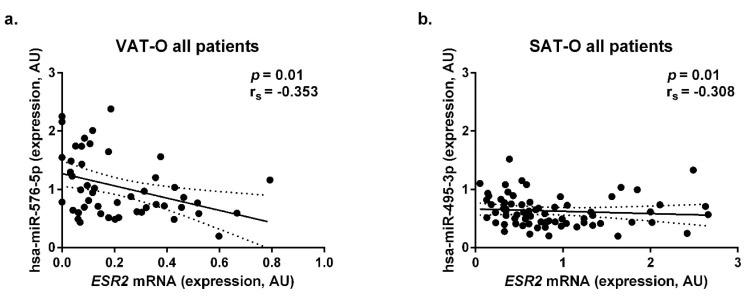
Correlations of *ESR2* mRNA concentrations with hsa-miR-576-5p (**a**) levels in the visceral adipose tissue (VAT) and with hsa-miR-495-3p levels (**b**) in the subcutaneous adipose tissue (SAT) of all obese (O) study participants. AU—arbitrary units.

**Table 1 ijms-23-05989-t001:** Selected clinical and biochemical parameters of the study participants.

	Obese Individuals before Weight Loss (N = 78)		Obese Individuals after Weight Loss (N = 19)		Normal-Weight Controls (N = 31)	
Males/Females	13/65		4/15		12/19	
	Mean ± SD	Min–Max	Mean ± SD	Min–Max	Mean ± SD	Min–Max
Age (years)	41.64 ± 10.34	20–62	41.47 ± 10.27	28–67	45.76 ± 14.81	23–62
Weight (kg)	130.35 ± 20.77	98.0–198.60	76.11 ± 7.15	68–90	67.71 ± 11.23	50–90
BMI (kg/m^2^)	46.24 ± 5.48	40.2–59.52	27.2 ± 2.35	24.30–29.51	23.42 ± 1.66	20.07–24.95
Adipose tissue (% body mass)	47.20 ± 5.20	32.64–57.23	30.5 ± 3.35	24.8–34.05	–	–
Waist circumference (m)	1.23 ± 0.18	0.95–1.67	0.90 ± 0.12	0.78–1.05	–	–
Weight loss (kg)	–	–	47.8 ± 10.4	35.2–65.6	–	–
TSH (mIU/L)	1.83 ± 1.22	0.33–5.63	1.58 ± 0.28	1.09–1.35	1.22 ± 0.18	1.09–1.35
Glucose (mmol/L)	5.98 ± 1.45	3.50–11.05	4.82 ± 0.55	4.12–5.66	5.21 ± 0.22	4.22–5.44
Total cholesterol (mmol/L)	4.91 ± 1.05	2.38–7.87	4.61 ± 0.88	3.52–5.89	4.85 ± 0.20	3.8–4.92
	**Obesity-Related Co-Morbidities**					
	N	%	N	%	N	%
Type 2 diabetes/prediabetes *	27	34.6	3	15.8	None	None
Hypertension	43	55.1	6	31.6	None	None
Hyperlipidemia	46	59.0	6	31.6	None	None
Metabolic syndrome **	41	52.5	None	None	None	None

BMI—body mass index calculated as weight (kg) divided by height squared (m^2^); CRP—C-reactive protein; HDL—high-density lipoproteins; LDL—low-density lipoproteins; N—number of subjects; TSH—thyroid-stimulating hormone; * impaired fasting glucose and/or impaired glucose tolerance; ** the metabolic syndrome was diagnosed based on the International Diabetes Federation criteria for Europeans [63].

## Data Availability

The data are in the possession of the corresponding author and can be provided upon request.

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
