# Peer review of "Epigenetic Regulation of Estrogen Receptor Genes’ Expressions in Adipose Tissue in the Course of Obesity"

_ijms, 2022, doi:10.3390/ijms23115989_

Round 1
Reviewer 1 Report
This manuscript shows Epigenetic regulation of estrogen receptors genes’ expression in adipose tissue in the course of obesity, which will be useful as a reference for future studies.
The data are solid and well-conducted and the manuscript is properly written.
Author Response
Reviewer 1
“This manuscript shows Epigenetic regulation of estrogen receptors genes' expression in adipose tissue in the course of obesity, which will be useful as a reference for future studies. The data are solid and well-conducted, and the manuscript is properly written.”
We thank the Reviewer for the positive assessment of our manuscript.

Reviewer 2 Report
In this manuscript, the authors investigated ESR1 and ESR2 expression in adipose tissues of obese individuals before and after weight loss and verifying whether epigenetic mechanisms are involved in this phenomenon. ESR1 and ESR2 mRNA and miRNA levels were evaluated by real-time PCR in visceral (VAT) and subcutaneous adipose tissue (SAT) of obese and normal-weight individuals and in SAT samples from the post-bariatric patients. ESR1 and ESR2 methylation status was studied using the methylation-sensitive digestion/real-time PCR method. These authors showed that obesity was associated with a decrease in mRNA levels of both ERs in SAT and ESR2 in VAT, while weight loss increased ESR transcription. Methylation levels of ESR1 and ESR2 promoters were unaffected. However, ESR1 mRNA in AT of obese subjects correlated negatively with the expression of several miRNAs. They conclude that obesity-associated downregulation of ER mRNA levels in adipose tissue may result from the miRNA interference. This study is of interest to the field because it is well known that estrogen receptors are involved in adipogenesis. My main concerns are:
- All the results are descriptive. The molecular mechanism was not attempted to explored.
- The gene expression level was solely assessed by real-time PCR , that is less persuasive for the alteration of ESR expression in the whole adipose tissue. it is better to be supported by assessing protein levels, or ESR-regulated adipose-target genes .
- There is no evidence to show the miRNAs examined could be potentially interact or associated with ESRs in adipose tissue.
Reviewer 3 Report
The authors investigate the expression of estrogen receptors a and B (ESR1 and ESR2), relevant miRNAs, and promoter methylation status in a valuable set of adipose tissue samples from obese and normal weight patients, plus post-weight-loss samples. They observe decreased ESR1 and ESR2 in samples from obese individuals in a depot-specific manner (SAT vs VAT). They begin to explore mechanisms associated with this decrease in gene expression, such as increased abundance of miRNAs that target these ER transcripts, and present the correlations for a variety of sample subgroups. The manuscript is very well-written, and the authors provide a comprehensive review of the literature that helps provide context for their study. My only comments are very minor and hopefully easily addressed:
Could the authors please include a brief description of how the ESR1/ESR2 ratio was calculated? Reference #65 describes the RT-PCR method and calculation by 2^-delta Ct, but could the authors briefly mention how the ESR1/ESR2 ratio was calculated? In Figure 3, is it likely that ESR1 is driving these changes in ER ratio (overall patterns tend to be more like ESR1)? Could the authors please comment on this?
Related to Figures 5 and 6: Are there correlations between ESR mRNA expression and select miRNA abundances in any non-obese tissue samples? Is this mode of regulation at work in healthy adipose tissue, or is this mechanism predominantly in expanded adipose tissue? My apologies if I missed this point in the discussion or elsewhere.
Line 589: reference #65 is cited for the miRNA quantitation, but I did not see those method details in that paper. Could the authors please provide more information about the miRNA quantitation method, including primer sequences, if available?
Is reference #9 cited in the manuscript?
Line 61: minor word choice: "mice of both sexes" instead of "mice of both genders"?
Line 78: minor word choice: instead of "prevalence" of obesity among rodent models, perhaps the word "propensity to develop"?
Lines 63-65 and 79-80: It seems like references 10 and 15 are cited similarly--is there a distinction between the use of these citations in the two sentences? Could these papers be cited together?
Lines 104, 116, 129, 307, 318: Instead of ESR1/2 "expression on mRNA level", consider using ESR1/2 "mRNA expression"
Line 115: When introducing the surgically-induced weight loss samples, would it be possible to explain how long after surgery these samples were collected here? (Or refer readers to the methods section where that information is provided?)
Line 233, 235, 249: TTS should be TSS for transcriptional start site
Line 264: Could the investigators please clarify if the "investigated tissues" from reference #26 are the same adipose depots as studied in this manuscript?
Line 268-269: Did the authors mean to include hsa-miR-450b-5p among the list of investigated miRNAs for ESR2, as in line 304?
Line 383-388: In discussing the link between adiposity and decreased ER expression here, do genetic models in mice help clarify this relationship and could any relevant studies be cited here?
Regarding the lack of changes in promoter methylation (Line 432-448), is there a potential role for the cellular heterogeneity of adipose tissue in masking changes in methylation? Do single-cell RNA-seq analyses of adipose provide any further clarity about which cell types express ER in the context of obesity? I'm not sure if these datasets exist, but could the authors please comment on any relevant analyses of cellular heterogeneity in adipose tissue? (This is also related to the brief discussion of macrophages later in the discussion.)
Thank you for sharing your well-written manuscript. It was a pleasure learning about your work on estrogen receptor expression in this unique set of adipose tissue samples.
Round 2
Reviewer 2 Report
All of my major concerns are properly addressed.